# Exploring Token Pruning in Vision State Space Models

**Zheng Zhan**[1*], **Zhenglun Kong**[12*], **Yifan Gong**[1], **Yushu Wu**[1], **Zichong Meng**[1]
**Hangyu Zheng**[3], **Xuan Shen**[1], **Stratis Ioannidis**[1], **Wei Niu**[3], **Pu Zhao**[1], **Yanzhi Wang**[1]
[1]Northeastern University, [2]Harvard University, [3]University of Georgia
{zhan.zhe, kong.zhe, yanz.wang}@northeastern.edu

## Abstract

State Space Models (SSMs) have the advantage of keeping linear computational complexity compared to attention modules in transformers, and have been applied to vision tasks as a new type of powerful vision foundation model. Inspired by the observations that the final prediction in vision transformers (ViTs) is only based on a subset of most informative tokens, we take the novel step of enhancing the efficiency of SSM-based vision models through token-based pruning. However, direct applications of existing token pruning techniques designed for ViTs fail to deliver good performance, even with extensive fine-tuning. To address this issue, we revisit the unique computational characteristics of SSMs and discover that naive application disrupts the sequential token positions. This insight motivates us to design a novel and general token pruning method specifically for SSM-based vision models. We first introduce a pruning-aware hidden state alignment method to stabilize the neighborhood of remaining tokens for performance enhancement. Besides, based on our detailed analysis, we propose a token importance evaluation method adapted for SSM models, to guide the token pruning. With efficient implementation and practical acceleration methods, our method brings actual speedup. Extensive experiments demonstrate that our approach can achieve significant computation reduction with minimal impact on performance across different tasks. Notably, we achieve 81.7% accuracy on ImageNet with a 41.6% reduction in the FLOPs for pruned PlainMamba-L3. Furthermore, our work provides deeper insights into understanding the behavior of SSM-based vision models for future research[2].

## 1 Introduction

Recent years have witnessed the rapid evolvement of the computer vision field in the era of deep learning. Significant research efforts have been devoted to designing effective and efficient architectures of deep neural networks (DNNs) for visual tasks. Convolution Neural Networks (CNNs) [29, 12, 23, 30] and Vision Transformers (ViTs) [6, 22, 31, 43] are two representative categories of backbone networks. Though ViTs exhibit superior modeling capabilities with the incorporation of the self-attention mechanism [6, 32], the complexity of self-attention grows quadratically as the input size increases. Inspired by the great potential of State Space Models (SSMs) for long sequence modeling with linear complexity in natural language processing (NLP) tasks [9, 24, 33, 44], the latest backbone network designs for visual tasks [10, 21] leverage SSM-based blocks. Particularly, VMamba [21] reduces the complexity of attention computation with the selective scan mechanism presented in the S6 model [9] and matches the performance with existing foundation models.

Like the existing research efforts promoting the efficiency of CNNs and ViTs, the exploration of the SSM efficiency is desirable to facilitate real-time applications. While weight pruning is the prevalent

---

*Equal contributions

[2]Code available at `https://github.com/ZLKong/ToP-ViM`

technique for CNNs [34, 13, 17, 14, 38, 7, 8, 35, 40], token pruning [27, 26, 39, 28, 16, 41, 5] proves to be an effective way to enhance the efficiency of ViTs due to the independent patch processing design. Given that the SSM-based blocks also process input by dividing it into patches like ViTs, the existing token pruning techniques [18] for ViTs can be applied as a straightforward approach to boost the SSM efficiency. However, as shown in Figure 2, although enjoying certain benefits of faster inference with less tokens, this naive token pruning application for SSMs suffers from significant accuracy drops. Even after extensive fine-tuning efforts, its accuracy is still not able to recover from the token pruning with non-marginal gaps compared with the original accuracy. This indicates that the direct application of token pruning designed for ViTs permanently harms the performance of SSM-based vision models.

Given this observation, we conduct a thorough analysis of the computation patterns in SSM-based blocks, aiming to find the root cause and provide a foundation for efficient token pruning design in SSMs. Unlike ViTs whose attention mechanism computes the correlation between each pair of patches, SSM-based blocks follow traversal paths and thus the paths are sensitive to their adjacent patches. The direct application of token pruning techniques from ViT disrupts the patch locations/neighborhood in SSM-based blocks, thus incurring massive accuracy drops.

Based on our analysis, the question naturally arises whether we can keep the sequential property of tokens/batches in SSM-based vision models while pruning tokens to accelerate the forward computation. A successful solution not only improves the computational efficiency, but also provides more insights into the interpretability of SSM scan/token for future research. We take the first novel step towards this direction by proposing a general token pruning method for SSM-based vision models. Specifically, we propose a token importance evaluation method adapted for SSM models to guide the token pruning process based on a comprehensive analysis of SSM-based models. More importantly, to address the root cause of the above significant accuracy drop, we introduce a pruning-aware hidden state alignment method to reform the scan mechanism in SSMs for pruned and remaining tokens, thus stabilizing the neighborhood of remaining tokens and enhancing performance. Following the token pruning designs, we explore the efficient implementation and practical acceleration methods. With our tailored design, the computations can be significantly reduced with high accuracy performance. Notably, we achieve 81.7% accuracy on ImageNet for token pruned PlainMamba-L3, with 41.4% FLOPs reduction. We summarize our contributions as follows:

- After observing the incapability of directly applying token-based pruning techniques from ViTs for vision SSMs, we conduct a comprehensive analysis of SSM-based blocks to identify the failure reason, as well as provide more insights for the SSM scan mechanism in vision tasks, shedding lights on future research on SSM-based vision models.

- Based on our analysis, we propose a general token pruning method for SSM-based vision models, incorporating an adapted token importance evaluation to determine the pruned tokens, a pruning-aware hidden state alignment method to reform the SSM scan mechanism for pruned and remaining tokens, and practical implementation for efficient inference.

- We take the first step towards accelerating vision SSM models with token-based pruning. Our extensive and comprehensive experiments for image classification and object detection demonstrate the effectiveness of our proposed method for vision SSMs.

## 2   Related Work

**State Space Models.**   SSMs [9, 24, 33] were first proposed to tackle long sequence modeling in the NLP community. The design has the strength to model complex systems by focusing on how the input, output, and state variables evolve over time. Recent progress has demonstrated that the variants of SSMs can be applied to visual tasks as an alternative to CNNs and ViTs with promising results. S4ND [25] is the first work that applies the state space mechanism to visual tasks and shows the potential to achieve competitive performance with ViTs [6]. The design expands the S4 model [10] and normalizes the parameters into a diagonal structure. But it fails to efficiently capture image information in an input-dependent manner. ViM [46] proposes a novel vision backbone with bidirectional Mamba. Based on that, PlainMamba [37] invents a continuous 2D scanning to enhance spatial continuity by ensuring adjacency of tokens in the scanning sequence. VMamba [21] introduce Cross-Scan Module (CSM) to enable 1D selective scan, matching the performance with existing foundation models including ResNet [12], ViT [6], Swin [22], and ConvNext [23]. The

great accomplishments demonstrate the potential of vision SSMs as an emerging fantastic foundation model family.

**Token Pruning.** Token pruning is an effective strategy to enhance computational efficiency by reducing the number of processed tokens or patches. It enables significant acceleration without requiring additional weights or specialized hardware, aiming to selectively retain the most informative tokens and sparsify the sequence. It is also vital for dense prediction tasks where sequence sizes are extensive. Several innovative approaches have been developed for vision transformers. For example, EViT [18] uses the attentiveness of the [CLS] token with respect to other tokens to identify the most important tokens. DynamicViT [27] and SPViT [15] add layers that employ the Gumbel-Softmax trick to selectively prune less informative tokens. IA-RED2 [26] drops redundant tokens with a multi-head interpreter. PS-ViT (T2T) [39] discard useless patches in a top-down paradigm. PATCHMERGER [28] uses spatial attention to generate a small set of tokens adaptive to the input. ToMe [2] measures dot product similarity between token keys to determine redundancy and prune accordingly. However, the dynamics of information flow between tokens and the learning mechanisms in models like Mamba [9] remain largely unexplored. Unlike ViTs that reply on attention features, the absence of attention layers in Mamba makes current pruning methods ineffective. Furthermore, the inclusion of the SSM module prevents the effective use of existing token pruning methods [42].

## 3 Preliminary and Motivation

### 3.1 State Space Models

State Space Models (SSMs) are sequential models that map an input sequence $x(t) \in \mathbb{R}^L$ to an output sequence $y(t) \in \mathbb{R}^L$ through a hidden state $h(t) \in \mathbb{R}^N$ as follows,

$$
\begin{aligned}
h'(t) &= \mathbf{A}h(t) + \mathbf{B}x(t), \\
y(t) &= \mathbf{C}h(t),
\end{aligned}
\tag{1}
$$

where $L$ denotes the length of the sequence, $N$ denotes the number of representation dimensions, $\mathbf{A} \in \mathbb{R}^{N \times N}$ is the evolution matrix, $\mathbf{B} \in \mathbb{R}^{N \times L}$, and $\mathbf{C} \in \mathbb{R}^{L \times N}$ are the projection matrices.

The Mamba model [9] represents a discrete version of the continuous system for SSMs and incorporates a timescale parameter $\Delta$ to facilitate the transformation of continuous parameters with the zero-order hold (ZOH) as follows,

$$
\begin{aligned}
\overline{\mathbf{A}} &= \exp(\Delta\mathbf{A}), \\
\overline{\mathbf{B}} &= (\Delta\mathbf{A})^{-1}(\exp(\Delta\mathbf{A}) - \mathbf{I}) \cdot \Delta\mathbf{B}.
\end{aligned}
\tag{2}
$$

After obtaining the discretized $\overline{\mathbf{A}}$ and $\overline{\mathbf{B}}$, the discretization of Equation (1) can be rewritten as follows,

$$
\begin{aligned}
h_t &= \overline{\mathbf{A}}h_{t-1} + \overline{\mathbf{B}}x_t, \\
y_t &= \mathbf{C}h_t.
\end{aligned}
\tag{3}
$$

Finally, the Mamba model computes the output through a global convolution as follows,

$$
\begin{aligned}
\overline{\mathbf{K}} &= (\mathbf{C}\overline{\mathbf{B}}, \mathbf{C}\overline{\mathbf{A}}\overline{\mathbf{B}}, \ldots, \mathbf{C}\overline{\mathbf{A}}^{\mathbf{L-1}}\overline{\mathbf{B}}), \\
\mathbf{y} &= \mathbf{x} * \overline{\mathbf{K}},
\end{aligned}
\tag{4}
$$

where $\mathbf{y}$ denotes the output sequence, $L$ denotes the length of the input sequence $\mathbf{x}$, and $\overline{\mathbf{K}} \in \mathbb{R}^L$ denotes a structured convolutional kernel.

### 3.2 Failure of Applying ViT Token Pruning for ViMs

> **(Observation)** After applying token pruning method to an SSM-based vision model, the **Zero-shot** performance will **drop significantly**. Moreover, this process will **permanently harm** the model's performance, even after **extensive fine-tuning**.

**Epic failure of traditional token pruning for vision SSMs.** To explore the token sparsity in vision SSMs, we first prune tokens in SSM-based models with the 'must-try' baseline, which directly applies the token pruning techniques designed for ViTs. Specifically, we prune the tokens using

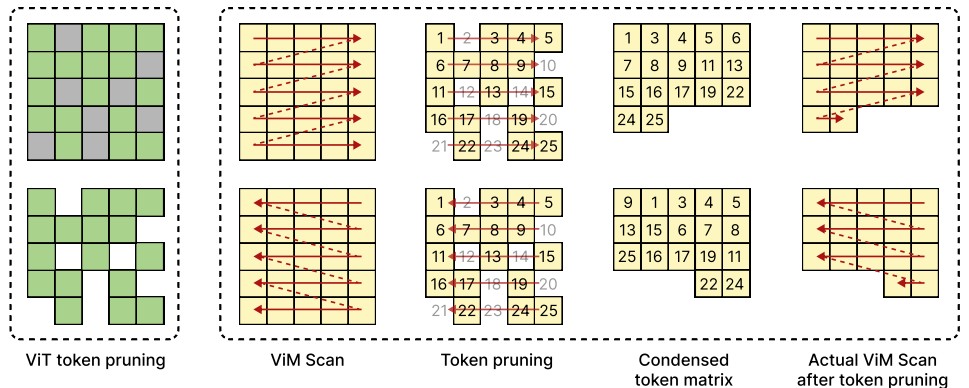

Figure 1: Illustration of the cross-scan in ViM models before and after token pruning.

selector metrics of EViT [18] on both transformer-based ViT-S [6] and SSM-based ViM-S models [46]. Given $N$ input tokens for one layer, with token pruning, $K$ tokens remain while the other $N - K$ tokens are pruned. The remaining tokens are relabeled as $\{x_j\}_{j=0}^{K-1}$ and their hidden states are obtained following Equation (3). In this way, the number of the tokens and the corresponding hidden states are reduced, condensing token matrix to save computation costs, as shown in Figure 1. After pruning tokens based on the token selector, we evaluate the performance in terms of zero-shot and fine-tuning accuracy. The results are shown in Figure 2. As observed, the direct application of token pruning from ViTs is not capable of delivering satisfying performance on ViMs. Specifically, direct token pruning suffers from substantial zero-shot accuracy degradation on ViM-S (with over 68% accuracy drop compared with the original accuracy), despite its success for ViT-S with merely 1.4% accuracy drop. Furthermore, even after extensive fine-tuning for the pruned model, its accuracy is not restored still with a 5.7% accuracy gap compared with the original ViM-S model, while it can boost the accuracy to a competitive level on ViT-S after fine-tuning. The significant performance degradation in ViM-S demonstrates that the direct application of token pruning hurts the underlying computations in SSM-based blocks, with permanent negative effects which can hardly be restored after fine-tuning.

**Computation patterns in vision SSM.** Observing the great success for ViT-S and epic failure for ViM-S with the same method, we are motivated to revisit the unique computation characteristics of SSMs and rethink the token pruning strategy in ViMs. To figure out the reason of failure, we look into the token computation patterns in SSM-based blocks. Given the input data, SSM-based blocks first unfold image patches/tokens into sequences along traversal paths (i.e., cross-scan, as shown in Figure 1 with ViM scan), process each token sequence using a separate computa-

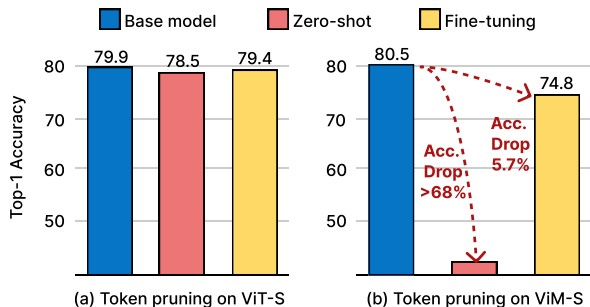

Figure 2: Accuracy comparison for token pruning on transformer-based ViT-S and SSM-based ViM-S.

tion block in parallel, and subsequently reshape and merge the resultant sequences to form the output map (i.e., cross-merge). The traversal paths facilitate the integration of information from all image pixels in various directions with linear complexity, enhancing the model's understanding.

**Reason for the failure.** However, the unique traversing along the sequence paths in ViM makes each token sensitive to its neighboring tokens. This is not a problem for ViT as the quadratic design of the attention mechanism calculates the correlation between the target token and all other tokens in the image, eliminating the sensitivity to adjacent tokens. As shown in Figure 1, introducing a token pruning strategy within an SSM-based block disrupts the original token positions in the SSM scan. Consequently, tokens that were not previously adjacent become neighbors during the scan in different directions or paths, leading to a distorted scan functionality and a significant accuracy degradation. Especially considering that the tokens are actually image patches in visual tasks with semantic information, disrupting their positions during the scan brings great difficulties to understand their relationship and the overall semantics.

In response to the limitations of directly applying existing token pruning methods designed for ViTs, we aim to address the following question:

> **(Question)** *Can we prune tokens in SSM-based vision models to accelerate their forward computation without disrupting the original sequential token positions in different directions during the scan?*

## 4  Methodology

To address the above **Question**, we propose a general token pruning method tailored for SSM-based vision models. Specifically, we propose a pruning-aware hidden state alignment method to stabilize the neighborhood of remaining tokens during the scan, addressing the distorted scan functionality in traditional token pruning and thus enhancing accuracy performance. Furthermore, based on our detailed analysis of SSM-based vision models, we propose a token importance evaluation method adapted for SSM models, to guide the token pruning. Moreover, we discuss the efficient implementation and practical acceleration methods for token-pruned SSM-based vision models.

### 4.1  Pruning-Aware Hidden State Alignment

To maintain the sequential property of SSM tokens during the scan and tackle the **Question**, we propose the following novel pruning-aware hidden state alignment technique to align the sequential positions or neighbourhood of tokens before and after token pruning during the scan, thus maintaining the model performance under token pruning. For SSM-based vision models, the input token sequence for the $l^{th}$ layer is denoted as $T_{l-1} \in \mathbb{R}^{B \times N \times D}$, where $B$, $N$, and $D$ are the batch size, token number, and hidden state dimension, respectively. The tokens in one batch of the sequence can be unfolded as $\{x_j\}_{j=0}^{N-1}$ with $N$ tokens in total. After applying token pruning (Section 4.2), $K$ tokens are kept while the other $N - K$ tokens are removed from the input token sequence. We adopt different strategies to align the hidden states of remained tokens and pruned tokens during the scan as detailed below.

**Alignment of hidden states for remaining tokens.** We denote the set of the remaining token indices as $\{q_j\}_{j=0}^{K-1}$ with $K$ elements and $q_s < q_t$ if $s < t$. Formally, the pruning-aware hidden states during the scan corresponding to the remained tokens can be represented as

$$
\begin{aligned}
h'_{q_0} &= \overline{\mathbf{B}} x_{q_0}, \\
h'_{q_1} &= \overline{\mathbf{A}}^{q_1 - q_0} \overline{\mathbf{B}} x_{q_0} + \overline{\mathbf{B}} x_{q_1}, \\
&\quad \dots \\
h'_{q_{(K-1)}} &= \underbrace{\overline{\mathbf{A}}^{q_{(K-1)} - q_0} \overline{\mathbf{B}} x_{q_0} + \overline{\mathbf{A}}^{q_{(K-1)} - q_1} \overline{\mathbf{B}} x_{q_1} + \overline{\mathbf{A}}^{q_{(K-1)} - q_2} \overline{\mathbf{B}} x_{q_2} + \dots + \overline{\mathbf{B}} x_{q_{(K-1)}}}_{K \text{ terms/tokens}}.
\end{aligned}
\tag{5}
$$

As shown in Equation (5), the hidden states of remained tokens depend on its current token and all previous remaining tokens. The pruned tokens are not effective in the hidden states.

**Alignment of hidden states for pruned tokens.** As observed in Figure 1 and 2, if one token is pruned, removing its position during the scan disrupts the neighbourhood of its adjacent tokens, leading to significant zero-shot accuracy drop which can hardly be compensated even after extensive fine-tuning. To mitigate this problem, our pruning-aware hidden state alignment maintains the position gap from pruned tokens during the scan to stabilize the neighbourhood of all remaining tokens. Specifically, to make the problem tractable, for two adjacent remaining tokens $x_{q_i}$ and $x_{q_{i+1}}$, if $q_{i+1} - q_i > 1$, meaning there are tokens pruned between $x_{q_i}$ and $x_{q_{i+1}}$, we denote the number of pruned tokens between $x_{q_i}$ and $x_{q_{i+1}}$ as $K_i$ ($K_i \geq 1$) and their indices can be represented as $\{q_i + j\}_{j=1}^{K_i}$. We have $q_i < (q_i) + 1 < \dots < (q_i) + K_i < q_{(i+1)}$. To highlight the difference between $q_{i+1}$ and $q_i + 1$, we use round brackets in the expression (e.g., $q_{(i+1)}$ and $(q_i) + 1$) without changing their meanings. Thus, the hidden states for the pruned tokens between two remaining adjacent tokens can be represented as follows,

$$h'_{q_i} = \overline{\mathbf{A}}^{q_i-q_0}\overline{\mathbf{B}}x_{q_0} + \overline{\mathbf{A}}^{q_i-q_1}\overline{\mathbf{B}}x_{q_1} + ... + \overline{\mathbf{B}}x_{q_i},$$

$$h'_{(q_i)+1} = \overline{\mathbf{A}}^{(q_i)+1-(q_0)}\overline{\mathbf{B}}x_{q_0} + \overline{\mathbf{A}}^{(q_i)+1-(q_1)}\overline{\mathbf{B}}x_{q_1} + ... + \overline{\mathbf{A}}\overline{\mathbf{B}}x_{q_i},$$

$$...\tag{6}$$

$$h'_{(q_i)+K_i} = \overline{\mathbf{A}}^{(q_i)+K_i-(q_0)}\overline{\mathbf{B}}x_{q_0} + \overline{\mathbf{A}}^{(q_i)+K_i-(q_1)}\overline{\mathbf{B}}x_{q_1} + ... + \overline{\mathbf{A}}^{K_i}\overline{\mathbf{B}}x_{q_i},$$

$$h'_{q_{(i+1)}} = \overline{\mathbf{A}}^{q_{(i+1)}-q_0}\overline{\mathbf{B}}x_{q_0} + \overline{\mathbf{A}}^{q_{(i+1)}-q_1}\overline{\mathbf{B}}x_{q_1} + ... + \overline{\mathbf{A}}^{q_{(i+1)}-q_i}\overline{\mathbf{B}}x_{q_i} + \overline{\mathbf{B}}x_{q_{(i+1)}}.$$

For pruned tokens with indices smaller than $q_0$, their hidden states are set to zero. For pruned tokens with indices larger than $q_{K-1}$, their hidden states can still be obtained following Equation (6). As shown in Equation (6), if a token is pruned, we do not simply remove its corresponding hidden state during the scan as it leads to substantial accuracy degradation shown in Figure 1 and 2. Instead, its hidden state in the scan can be obtained by using the previous state with one step forward, i.e., $h'_{(q_i)+1} = \overline{\mathbf{A}}h'_{(q_i)} + \overline{\mathbf{B}}x_{(q_i)+1} = \overline{\mathbf{A}}h'_{(q_i)}$ where the token $x_{(q_i)+1}$ is pruned. In this way, the hidden states corresponding to pruned tokens are aligned with that of the original unpruned tokens to maintain the sequential positions of the original tokens without disrupting their neighbours.

**Comparison with traditional token pruning.** As discussed in Section 3.2, in traditional ViT token pruning, the remaining tokens are relabeled as $\{x_j\}_{j=0}^{K-1}$ (disrupting their neighbours due to removal of pruned indices) and their hidden states are obtained following Equation (3). Different from ViT token pruning, we still keep the original indices of all tokens (including remaining and pruned tokens) to record their original sequential positions and neighbourhood. During the scan, the hidden states of pruning tokens becomes completely zero in ViT token pruning, which is different from our adapted scan mechanism in Equation (6) to keep a copy from its previous unpruned neighbour.

## 4.2 Token Pruning based on Importance Evaluation

In SSM-based vision models such as ViM, for the $l^{th}$ layer, the input token sequence $T_{l-1} \in \mathbb{R}^{B \times N \times D}$ is first projected to $X' \in \mathbb{R}^{B \times N \times D'}$, and then goes through bidirectional SSMs for data-dependent global visual context modeling. It processes $X'$ from the `forward` and `backward` scan via:

$$y_m \leftarrow \mathtt{SSM}(A_m, B_m, C_m)(X'_m), \quad \text{for } m \in \{\mathtt{forward}, \mathtt{backward}\},\tag{7}$$

where $y_m \in \mathbb{R}^{B \times N \times D'}$ is the output of SSM. Then $y_m$ is gated to obtain $y'_{\mathtt{forward}}$ and $y'_{\mathtt{backward}}$. The token sequence output of the $l^{th}$ layer can be obtained as follows:

$$T_l \leftarrow \mathtt{Linear}^T(y'_{\mathtt{forward}} + y'_{\mathtt{backward}}) + T_{l-1}.\tag{8}$$

Therefore, the output of SSM can directly reflect the token importance. The Mamba architecture, with its high-dimensional channel space, allows for a finer-granularity analysis of attention across numerous channels. Unlike Transformers that produce a single attention matrix per head, Mamba models exploit their extensive channel capacity for a more detailed attention distribution, enhancing the model's ability to discern subtle features and interactions among tokens. Thus, we aggregate the clipped values across all channels for each token to evaluate token importance as follows,

$$\mathcal{S} = \frac{\sum_{d=1}^{D'} \mathtt{max}(0, [\mathbf{y}]_{::d})}{D'},\tag{9}$$

where $[\cdot]_{::d}$ denotes the $d^{th}$ feature map in the feature dimension with size $D'$. We use $\mathcal{S}$ as the token importance metric to guide the pruning process, ensuring that only the most contextually relevant tokens are retained, thereby optimizing computational resources. Given the sparsity requirement for the token pruning, we sort $\mathcal{S}$ and prune the corresponding less important tokens. To make a comprehensive study, we compare the performance with other token importance metrics, including the $\ell_1$ norm, $\ell_2$ norm, as well as unclipped values without the `max` operation. We find that using clipped values in Equation (9) as the token importance metric can constantly yield better results.

### 4.3 Efficient Implementation and Practical Acceleration

**Efficient implementation for the SSM scan.** Based on the pruning-aware hidden state alignment technique discussed in Section 4.1, we propose the `pruning-aware hidden state alignment` kernel for practical acceleration. It utilizes a position map to guide the SSM operator, ensuring the correctness of computations. The position map is the token pruning indicator based on $\{q_j\}_{j=0}^{K-1}$ in Section 4.1, which inherits from token importance evaluation and records the location of remained tokens and pruned tokens. The `pruning-aware hidden state alignment` kernel takes the pruned dense sequences and the position map as its inputs. During the scan, it switches between the token remaining pattern and the token pruned pattern based on the remaining/pruning state of the token indexed by the position map. The token remaining pattern in the kernel follows the computations in Equation (5). Similarly, following Equation (6), the token pruned pattern still updates the hidden states but ignores computations related to the current token. The kernel switches to another pattern if it detects a corresponding change in token pruning state. A pseudo-code for our `pruning-aware hidden state alignment` is demonstrated in Appendix A. Thus, the `pruning-aware hidden state alignment` kernel can effectively accelerate the SSM scan under token pruning.

**Practical acceleration for the whole model.** Note that the SSM scan only takes up around 10∼20% computations in the whole model. With less tokens, other parts in the model can be accelerated directly due to less computations from pruned tokens, leading to significant inference speedup performance as demonstrated in our experiments.

## 5 Experiment Results

We conduct comprehensive experiments on ImageNet-1K[4], COCO 2017 [20] and ADE20K [45] datasets. All experiments are conducted on 4 NVIDIA V100s. "-EViT" means apply EViT token pruning method. "-prune" means apply our token pruning method. We report average results of multiple runs for all experimental sections, and different runs do not vary much. For ViM-T, we prune after the 10th and 20th layers. For ViM-S, we prune after the 5th, 10th, 15th, and 20th layers. For PlainMamba-L1, we prune after the 5th and 10th layers. For PlainMamba-L2, we prune after the 5th, 10th and 15th layers. For PlainMamba-L3, we prune after the 5th, 11th, 17th, and 23th layers.

### 5.1 Image Classification on ImageNet-1K

**Settings.** We finetune both the ViM and PlainMamba for 30 epochs on the ImageNet-1K dataset. The top-1 accuracy on the validation set is reported. For ViM, we set a patch extraction stride of 8 while keeping the patch size unchanged, a constant learning rate of $10^{-5}$, and a weight decay of $10^{-8}$. For PlainMamba, we use a warm-up period of 5 epochs. The weight decay is set to 1e-8, the base learning rate to 2e-5, the warm-up learning rate to 2e-8, and the minimum learning rate to 2e-7.

**Results.** The comparison results of our token pruning models against benchmark backbone models on ImageNet-1K are summarized in Table 1. One advantage of our method is that it is general and can be applied to a wide range of SSM-based vision model architectures to reduce computation complexity with a minor loss of performance. We evaluate our method on five base models including ViM-T, ViM-S, PlainMamba-L1, PlainMamba-L2, and PlainMamba-L3. We report the top-1 accuracy and FLOPs. Compared to directly applying the EViT method on vision state space models, using our pruning-aware hidden state alignment and token importance metric constantly outperforms EViT across various models of different scales. Specifically, On ViM, our method surpasses EViT by 3.8% on ViM-T and 4.0% on ViM-S. On PlainMamba, our method exceed EViT by 2.4% on PlainMamba-L1, 2.7% on PlainMamba-L2, and 2.8% on PlainMamba-L3.

### 5.2 Object Detection and Instance Segmentation

**Settings.** Following previous works, we conduct experiments for object detection and instance segmentation on the COCO 2017 dataset. The COCO 2017 dataset contains 118K images for training, 5K images for validating, and 20K images for testing. We use both the two-stage Mask R-CNN [11] and the single-stage RetinaNet [19]. For both models, we report the results of both 1× schedule. Following [37], we use ViTAdapter [3] to compute multi-scale features to fit the FPN network structure.

Table 1: Classification results of different models on ImageNet-1K. We compare the proposed token pruning method with existing methods under comparable GFLOPs.

| Method | Img. Size | Params (M) | FLOPs(G) | Top-1 Acc. (%) |
|---|---|---|---|---|
| ViT-Base | $384^2$ | 86 | 55.40 | 77.9 |
| ViT-Large | $384^2$ | 307 | 190.70 | 76.5 |
| DeiT-Tiny | $224^2$ | 6 | 1.30 | 72.2 |
| DeiT-Small | $224^2$ | 22 | 4.60 | 79.8 |
| DeiT-Base | $224^2$ | 86 | 17.50 | 81.8 |
| ViM-T | $224^2$ | 7 | 1.50 | 76.1 |
| ViM-S | $224^2$ | 26 | 5.10 | 80.5 |
| ViM-T-EViT | $224^2$ | 7 | 1.28 (-14.3%) | 71.3 |
| ViM-S-EViT | $224^2$ | 26 | 3.57 (-30.0%) | 74.8 |
| ViM-T-ToP | $224^2$ | 7 | 1.29 (-14.0%) | 75.1 |
| ViM-S-ToP | $224^2$ | 26 | 3.60 (-29.4%) | 78.8 |
| PlainMamba-L1 | $224^2$ | 7 | 3.0 | 77.9 |
| PlainMamba-L2 | $224^2$ | 25 | 8.1 | 81.6 |
| PlainMamba-L3 | $224^2$ | 50 | 14.4 | 82.3 |
| PlainMamba-L1-EViT | $224^2$ | 7 | 2.44 (-18.7%) | 75.0 |
| PlainMamba-L2-EViT | $224^2$ | 25 | 6.22 (-23.2%) | 78.3 |
| PlainMamba-L3-EViT | $224^2$ | 50 | 8.35 (-42.0%) | 78.9 |
| PlainMamba-L1-ToP | $224^2$ | 7 | 2.46 (-18.0%) | 77.4 |
| PlainMamba-L2-ToP | $224^2$ | 25 | 6.27 (-22.6%) | 81.0 |
| PlainMamba-L3-ToP | $224^2$ | 50 | 8.44 (-41.4%) | 81.7 |

Table 2: Results on COCO object detection and instance segmentation.

| Backbone | $AP^b$ | $AP^b_{50}$ | $AP^b_{75}$ | $AP^m$ | $AP^m_{50}$ | $AP^m_{75}$ |
|---|---|---|---|---|---|---|
| PVT-Small | 40.4 | 62.9 | 43.8 | 37.8 | 60.1 | 40.3 |
| PVT-Medium | 42.0 | 64.4 | 45.6 | 39.0 | 61.6 | 42.1 |
| PVT-Large | 42.9 | 65.0 | 46.6 | 39.5 | 61.9 | 42.5 |
| Swin-Tiny | 42.7 | 65.2 | 46.8 | 39.3 | 62.2 | 42.2 |
| Swin-Small | 44.8 | 66.6 | 48.9 | 40.9 | 63.2 | 44.2 |
| PlainMamba-L1 | 44.1 | 64.8 | 47.9 | 39.1 | 61.6 | 41.9 |
| PlainMamba-L2 | 46.0 | 66.9 | 50.1 | 40.6 | 63.8 | 43.6 |
| PlainMamba-L3 | 46.8 | 68.0 | 51.1 | 41.2 | 64.7 | 43.9 |
| PlainMamba-L1-EViT | 41.9 | 62.8 | 45.7 | 37.2 | 60.1 | 40.2 |
| PlainMamba-L2-EViT | 43.7 | 64.2 | 47.6 | 38.3 | 62.2 | 41.9 |
| PlainMamba-L3-EViT | 44.2 | 66.4 | 49.7 | 39.5 | 62.8 | 42.7 |
| PlainMamba-L1-ToP | 43.7 | 64.6 | 47.4 | 38.9 | 61.3 | 41.5 |
| PlainMamba-L2-ToP | 45.5 | 66.2 | 49.9 | 40.1 | 63.3 | 42.7 |
| PlainMamba-L3-ToP | 46.5 | 67.7 | 50.8 | 40.6 | 64.1 | 43.4 |

**Results.** We used our pruned PlainMamba models as the backbone and compared them with existing token pruning methods and dense backbones. As shown in Table 2, our token pruning method maintains similar performance to dense models (less than 0.5%). When compared to existing token pruning methods, specifically for PlainMamba-L1, our pruning method outperforms EViT-based pruning by an average of 1.59% across all six precision metrics. For PlainMamba-L2, our method surpasses EViT by an average of 1.63% across all six precision metrics. Additionally, for PlainMamba-L3, our method exceeds EViT by an average of 1.30% across all six precision metrics.

## 5.3 Semantic Segmentation on ADE20K

**Settings.** We conduct experiments for semantic segmentation on the ADE20K dataset [45]. ADE20K contains 150 fine-grained semantic categories, with 20K, 2K, and 3K images for training, validation,

and testing, respectively. We choose UperNet [36] as our base framework. We train all models for 160 iterations with batch size 16 and set the default training image size to 512×512.

**Results.** We show the results of semantic segmentation on ADE20K in Table 3. The results indicate that our method also works well for the semantic segmentation task by greatly reducing the computation costs while maintaining satisfying performance. For instance, our token pruned PlainMamba-L1 reaches a mIoU of 44.1%, which is the same as the unpruned PlainMamba-L1. Our PlainMamba-L3-prune has a mIoU of 48.6%, which is better than current state-of-the-art model architectures including LocalVim-S and VMamba-T.

Table 3: Semantic Segmentation.

| Method | mIoU/% |
|---|---|
| ViM-T | 41.0 |
| ViM-S | 44.9 |
| LocalVim-T | 43.4 |
| LocalVim-S | 46.4 |
| PlainMamba-L1 | 44.1 |
| PlainMamba-L2 | 46.8 |
| PlainMamba-L3 | 49.1 |
| PlainMamba-L1-EViT | 42.2 |
| PlainMamba-L2-EViT | 44.1 |
| PlainMamba-L3-EViT | 46.3 |
| PlainMamba-L1-ToP | 44.1 |
| PlainMamba-L2-ToP | 46.5 |
| PlainMamba-L3-ToP | 48.6 |

## 5.4 Ablation & Analysis

### 5.4.1 Token Importance Metric Analysis

In Table 4, we study on the impact of different token importance metrics, focusing on pruning-aware hidden state alignment. We test on two models: ViM-S and PlainMamba-L3. For the ViM-S model, both $\ell_1$-norm and $\ell_2$-norm methods achieve an accuracy of 78.6%, while the method without clipping (w/o Clip) results in a lower accuracy of 77.4%. The proposed clipping method (Clip) achieves the highest accuracy of 78.8%. For the L3 model, similar trends are observed: the $\ell_1$-norm and $\ell_2$-norm methods yield accuracies of 81.6% and 81.5%, respectively. The non-clipping approach results in a decrease in accuracy to 80.5%, whereas the clipping method provides a better enhancement, achieving 81.7%. These results suggest that the clipping mechanism in token importance metrics offers a consistent improvement in model accuracy, particularly in the context of pruning-aware hidden state alignment. It can potentially mitigate the adverse effects of extreme token importance values.

### 5.4.2 Quantitative Evaluation of pruning-aware hidden state alignment.

Table 5: Comparison of w/o and w/ our alignment (both using Eq. (9) as token importance metric).

| Model | Method | FLOPs | Top-1 Acc. (%) | Throughput |
|---|---|---|---|---|
| ViM-S | Dense | 5.10G | 80.5 | 1× |
| | Prune w/o our alignment | 3.57G | 75.4 | 1.30× |
| | **Prune w/ our alignment** | **3.60G** | **78.8** | 1.27× |
| PlainMamba-L3 | Dense | 14.40G | 82.3 | 1× |
| | Prune w/o our alignment | 8.35G | 79.3 | 1.47× |
| | **Prune w/ our alignment** | **8.44G** | **81.7** | 1.43× |

In Table 5, we compare the performance of different pruning methods across two models: ViM-S and PlainMamba-L3. Our pruning method without the alignment process reduces the FLOPs to 3.57G but also lowers the accuracy to 75.4%, resulting in an improved throughput of 1.30×. In contrast, adding the align matrix achieves a much higher accuracy of 78.8%, with a similar throughput of 1.27×. For the PlainMamba-L3 model, our pruning method without the alignment reduces FLOPs to 8.35G but decreases accuracy to 79.3%, while increasing throughput to 1.47×. Equipping the alignment process improves accuracy to 81.7% and achieves a throughput of 1.43×. These results demonstrate that the proposed pruning method with the alignment process can effectively balance computational efficiency and model accuracy, outperforming the baseline pruning approach.

Table 4: Ablation study of token importance metric with pruning-aware hidden state alignment .

| Model | Method | Accuracy (%) |
|---|---|---|
| ViM-S | $\ell_1$-norm | 78.6 |
| | $\ell_2$-norm | 78.6 |
| | w/o Clip | 77.4 |
| | Clip (ours) | 78.8 |
| L3 | $\ell_1$-norm | 81.6 |
| | $\ell_2$-norm | 81.5 |
| | w/o Clip | 80.5 |
| | Clip (ours) | 81.7 |

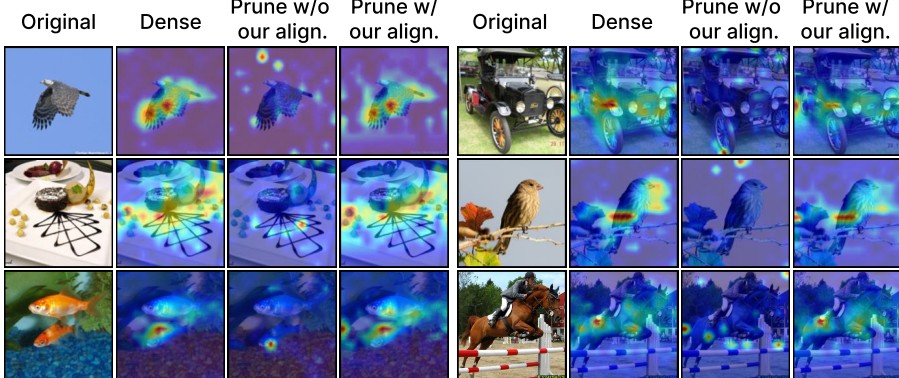

Figure 3: Visual representation on ImageNet-1K. We present the original images, attention visualizations from ViM-S, and **zero-shot results** of w/o and w/ our alignment method after the final layer.

## 5.5 Visualization and Interpretability

To further interpret token pruning in SSM-based vision models and understand the pruning-aware hidden state alignment behavior of our approach, we present attention visualizations based on zero-shot results in Figure 3. Our pruning-aware hidden state alignment effectively aligns the hidden states of pruned tokens in the SSM scan, maintaining similar visual representations and attention regions as the dense model. In contrast, pruning without our alignment shows significantly different attention regions, which could explain the huge accuracy drop. This demonstrates the effectiveness of our proposed pruning-aware hidden state alignment. The visualization tool is adopted from [1].

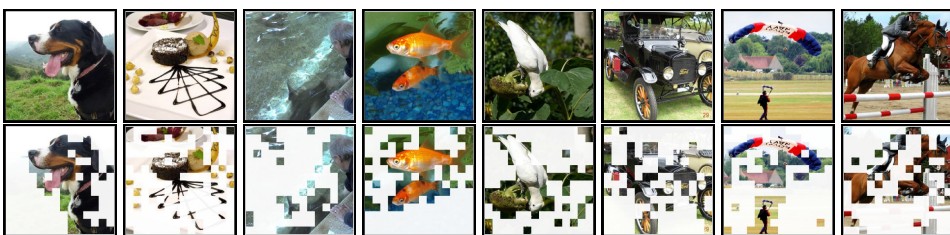

Figure 4: visualizations of locations of pruned token. We use the output after the final layer to visualize this reduction results.

We further visualize the token reduction results of our method within Fig. 4. We show the input images along with their sparsification results. The masked regions represent the tokens that have been pruned. Our method can gradually drop less informative tokens during forward pass and preserve the tokens that contain representative regions with an adaptive pruned region for each image.

## 6 Conclusion and Limitation

In this paper, we take the first step toward accelerating vision SSM models with token-based pruning. We analyze SSM-based blocks to understand the failure of direct token pruning and propose a general token pruning method for SSM-based vision models. This method includes an adapted token importance evaluation, a pruning-aware hidden state alignment, and practical implementations for efficient inference. Our extensive experiments confirm the effectiveness of our method and provide deeper insights into the SSM scan mechanism, guiding future research on SSM-based vision models. Though our method is general, the efficiency is limited by baseline model architecture design.

## Acknowledgement

This work is supported by National Science Foundation CNS-2312158, and also CCF-2428108, OAC-2403090. We would like to express our sincere gratitude to the reviewers for their invaluable feedback and constructive comments to improve the paper.

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

# Appendix

## A   Pseudo-code Example

---

**Algorithm 1:** PRUNING-AWARE HIDDEN STATE ALIGNMENT

---

```python
#example code of pruning aware hidden state alignment
def pruning_aware_hsa(state, position_map, x, dt, A, B, C, y_ptr):
    dA = exp(A * dt);
    if position_map:
        #remained token computation as Eq.5
        dB = B * dt;
        state = state * dA + dB * x;
        y_ptr = &sum(state * C);
    else:
        #pruned token computation as Eq.6
        state = state * dA;
        x.ptr++;
    return state
```

---

This is a pseudo-code example of our pruning-aware hidden state alignment for demonstration.

