# OpenReview forum: "Exploring Token Pruning in Vision State Space Models"
_NeurIPS.cc/2024/Conference — NeurIPS 2024 poster_

### Official Review · Reviewer_g8vG · 2024-07-03

**Soundness:** 3
**Presentation:** 3
**Contribution:** 2
**Rating:** 6
**Confidence:** 4

**Summary:**

The paper  introduces a pruning-aware hidden state alignment method, stabilizing token neighborhoods and maintaining model performance during token pruning. It also proposes an adapted token importance evaluation method tailored for SSM-based models, effectively guiding the pruning process. Extensive experiments demonstrate the method's efficacy, achieving significant computation reduction with minimal performance impact. Notably, the approach achieves 81.7% accuracy on ImageNet with a 41.6% reduction in FLOPs for pruned PlainMamba-L3, highlighting its practical acceleration benefits and effectiveness in maintaining high accuracy in vision tasks.

**Strengths:**

- the proposed method ensures the stability of token neighborhoods, maintaining model performance even after pruning.

- The importance evaluation is tailored specifically for SSM-based models, this evaluation method effectively guides the token pruning process.

- The approach achieves significant computation reduction with minimal impact on performance, exemplified by 81.7% accuracy on ImageNet with a 41.6% reduction in FLOPs for pruned PlainMamba-L3.

- The paper provides extensive visualization to further validate the effectiveness

**Weaknesses:**

- The proposed method is limited to plain, non-hierarchical SSM-based models.

**Questions:**

Can the proposed method generalize to the hierarchical variants like VMamba?

**Limitations:**

The authors adequately addressed the limitations

---

> ### Author Rebuttal · Authors · 2024-08-07
>
> We sincerely appreciate the reviewer for recognizing the strengths of our papers and providing valuable feedback. We are happy to address the raised questions as below.
>
> ---
> **W1. The proposed method is limited to plain, non-hierarchical SSM-based models.**
>
> We would like to kindly point out that ViM has proved to be the first fast and strong baseline for vision state space models and its design has been widely adopted by later vision state space models. That’s why we choose the ViM and its variants as our base models.
>
> Moreover, all the token reduction methods face similar difficulties when adapting to hierarchical model architecture due to their conflicts with the nature of hierarchical model architecture. However, with analysis and specific designs, the efficiency can still be achieved on hierarchical model architecture with our method. We discuss more design details in the following Response to **Q1**  to show how we can tackle this problem with our method.
>
> ---
> **Q1. Can the proposed method generalize to the hierarchical variants like VMamba?**
>
> Thanks for pointing this out, following the settings from previous token pruning work on Swin[1], we apply the sparsification operations at **stage 3** of Vmamba which contributes most of the complexity.  This is because the Layer number inside each stage of  VMamba-T is [2,2,8,2]  and  VMamba-S is [2,2,15,2], with major computations in Stage 3.
>
> |Methods | prune ratio at stage 3 | FLOPs(G) |  Top-1 Acc. (%)|
> |-|-|-|-|
> | VMamba-T| - |4.9|82.6|
> | VMamba-T-prune| 0.8| 4.0|82.1 |
> | VMamba-S | - |8.7|83.6|
> | VMamba-S-prune| 0.8|6.1 |83.2 |
>
> The toke pruning process cannot transfer across stages due to the downsampling process between stages. Therefore, we need to perform padding to restore the sequence at the end of the stage. But significant speedup inside stages can be achieved.
>
> [1] Dynamic Spatial Sparsification for Efficient Vision Transformers and Convolutional Neural Networks. TPAMI

---

> ### Author Response · Authors · 2024-08-12
>
> Dear Reviewer,
>
> Thank you very much for spending time reviewing our paper and acknowledging our contributions. Since the discussion will end very soon, we sincerely hope that you have found time to check our detailed response to your previous questions/comments. If you have any further questions, please feel free to let us know. We will try our best to reply to you before the discussion deadline.
>
> Thank you very much,
>
> Authors

---

### Official Review · Reviewer_TrFe · 2024-07-10

**Soundness:** 2
**Presentation:** 3
**Contribution:** 3
**Rating:** 5
**Confidence:** 3

**Summary:**

This manuscript aims to propose an effective pruning method for SSMs to achieve great trade-off of computation overhead and accuracy. The authors find that directly applying pruning strategies designed for transformer structure would greatly impair the performance of SSMs and give related analysis, that is SSMs take traversal path to consturct the interaction between different tokens and simple pruning would disrupt this connection. To address this challenge, the author first propose a strategy to build the alignment between the remaining tokens with the pruned tokens by keeping the indices and part calculation process of the pruned tokens. Based on the output of SSM layers, the author also propose a token selection strategy.

**Strengths:**

* The authors give a clear presentation of the motivation and the proposed method. The structure of the manuscript is well-organized.
* The motivation for designing specific pruning strategy for SSMs that take traversal interaction path is reasonable. The experiments of directly applying pruning method of EViT makes sense.
* Experiment results show that the proposed method reduces comparable FLOPs with directly applying EViT pruning strategy yet achieves better performance.

**Weaknesses:**

* The authors only compare the proposed method with EViT. However, it seems that EViT is proposed in 2022. The authors should consider comparing with more recent pruning methods.
* Lack of clear explanation and analysis of why the proposed pruning-aware hidden state alignment strategy is helpful. From Eq.6 we can see that the hidden state of h_{q+1} do not utilize any information of the pruned tokens, while the indices of evolution matrix A are changed. This part is confusing and I hope the author could give more descriptions about why only changing the indices brings such remakable improvements.
* The proposed Token Pruning based on Importance Evaluation does not make sense. Why the output of SSM can directly reflect the token importance? Just because it shares the same length with input? I recommend the authors could give move explanations.

**Questions:**

* I have an additional explanation about why directly applying pruning strategy of EViT greatly impair the performance of SSMs. Because transformer takes non-local operation, all the tokens have knowledge of the overall context, so that removing some tokens would not cause huge information loss. However, for SSMs, the traversal interaction path makes them greatly rely on the continuous context and token pruning would lead to huge information loss. If space available, I am happy to discuss with the authors about that.

**Limitations:**

The authors have discussed the potential limitations.

---

> ### Author Rebuttal · Authors · 2024-08-07
>
> We sincerely appreciate the reviewer for recognizing the strengths of our papers and providing valuable feedback. We are happy to address the raised questions as below.
>
> ---
> **W1. More comprehensive comparison.**
>
> We agree that a more comprehensive comparison would help the audience better understand the contribution of our work. We would also like to point out that EViT is still considered a strong baseline for token pruning methods. As for other token reduction methods, there have been recent advancements on Token merging as well as pruning + merge. Therefore, we implemented ToMe[1] as well as LTMP [2]  for our SSM-based models to provide a comprehensive comparison with state-of-the-art techniques. As demonstrated in the following table, our method can outperform all baselines with non-marginal improvements.
>
> |Methods | FLOPs(G) |  Top-1 Acc. (%)|
> |-|-|-|
> | ViM-T  |1.50| 76.1 |
> | ViM-T-ToMe |	1.28|71.6|
> | ViM-T-LTMP |	1.29|72.2|
> | ViM-T-EViT | 1.28|71.3|
> | ViM-T-Ours | 1.29 |75.1|
>
> [1] Token Merging: Your ViT But Faster, ICLR 2023
>
> [2] Learned Thresholds Token Merging and Pruning for Vision Transformers, TMLR 2023
>
> ---
> **W2. Proposed pruning-aware hidden state alignment strategy explanation.**
>
> As discussed in Section 3.2, for ViT, the hidden state of the pruned token is removed, i.e., $ h^\prime_{(q_{i})+1} = 0$ if the token $x_{(q_{i})+1}$ is pruned, which leads to substantial accuracy drop  as this strategy dropping hidden states  within an SSM block disrupts the original token positions in the SSM scan.  Consequently, tokens that were not previously adjacent become neighbors during the scan in different directions or paths, leading to a distorted scan functionality and a significant accuracy degradation.  Especially considering that the tokens are actually image patches in visual tasks with semantic information, disrupting their positions during the scan brings great difficulties to understand their relationship and the overall semantics.
>
> To address this issue, our method adopts $ h^\prime_{(q_{i})+1}  = \mathbf{\overline{A}} h^\prime_{(q_{i})} + \mathbf{\overline{B}} x_{(q_{i})+1} = \mathbf{\overline{A}} h^\prime_{(q_{i})}$ where the token $x_{(q_{i})+1}$ is pruned. It leads to Equation (6). As shown in Equation (6), if a token is pruned, we do not simply remove its corresponding hidden state during the scan as it leads to substantial accuracy degradation. Instead, its hidden state in the scan can be obtained by using the previous state with one step forward, i.e., $ h^\prime_{(q_{i})+1}  = \mathbf{\overline{A}} h^\prime_{(q_{i})}$. In this way, the hidden states in SSM corresponding to pruned tokens are aligned with that of the original unpruned tokens to maintain the sequential positions of the original tokens without disrupting their neighbours.  For the question “our method does not utilize any information of the pruned tokens”, our method is aware of the pruned tokens and can maintain the sequential information of the hidden states even if the tokens are pruned.
>
> We would like to clarify that in SSM models, the SSM scan keep doing a sequential computation by multiplying the previous hidden state with evolution matrix A and add the next hidden to the current state, Therefore, for the concern that **SSM did not change the indices of evolution matrix A**, the changing number on the head of evolution matrix A is the **exponent** of A. We maintain the position information by maintaining the sequential positions of the original tokens without disrupting their sequential relation of evolution matrix A.
>
> ---
> **W3. Explanation of our design of Importance Evaluation.**
>
> Please refer to Global rebuttal **A2** for more detailed explanation.
>
> Mamba utilizes implicit attention within its layers. It processes information through SSM layers, allowing tokens to interact and influence each other. By the time tokens reach the output of the SSM layers, they have undergone multiple rounds of implicit interactions and transformations. Mamba 2[1], has shown that the SSM block is equivalent to a form of linear attention. Therefore, the output contains the cumulative effect of these interactions, reflecting how each token has contributed to and been influenced by the overall context of the input. We then aggregate the clipped values across all channels of the output as our token importance score, as discussed in Line 228-231.
>
> In Figure 3 of our paper, we provide a visualization of the attention patterns derived from the output, which helps to illustrate the implicit attention mechanism in mamba.
> Moreover, deriving token importance directly from the model itself without additional token selection layers or algorithms can be beneficial for specific hardware optimization.
>
> [1] Transformers are SSMs: Generalized Models and Efficient Algorithms Through Structured State Space Duality.
>
> ---
> **Q1. Insight discussion.**
>
> We would like to first thank the reviewer for such an insightful discussion. In “Reason for the failure” discussion of section 3.2 of our paper, we try to find out the reason why token pruning can work on ViT while failing on ViM. An overall explanation could be that the non-local operation of transformers is due to the quadratic operation of self-attention’s token wise multiplication. This enables the model to gain comprehensive knowledge of the overall context.
>
> Moreover, our explanation on SSM aligns with the reviewer's thought on vision state space models that the SSM relies on sequential information (continuous context), directly removing the tokens could lead to huge information loss.
>
> Our results in Table 1 show that plainmamba could achieve larger FLOPs reduction (-41.4%) while maintaining good performance (-0.6%). This could be because plainmamba chose a continuous 2D scanning which scans from 4 directions, providing more overall context and robustness compared with 1D scanning in ViM, which lead to higher token pruning ratios.

---

> > ### Comment · Reviewer_TrFe · 2024-08-10
> >
> > Thanks for the authors' effort and rebuttal. Most of my concerns have been addressed. I'm leaning to accepting this manuscript.

---

> > > ### Author Response · Authors · 2024-08-10
> > >
> > > We sincerely thank the reviewer for recognizing that most concerns have been addressed and for leaning to accepting this manuscript. Given the enhancements made, we hope that our work now meets the criteria for a higher rating.
> > >
> > > We are truly grateful for your efforts in helping us refine our paper, and we look forward to your final assessment.

---

### Official Review · Reviewer_M2Lk · 2024-07-12

**Soundness:** 3
**Presentation:** 3
**Contribution:** 3
**Rating:** 6
**Confidence:** 4

**Summary:**

This paper introduces a token pruning method for vision SSMs to improve computational efficiency while maintaining performance. The authors identify that direct application of existing token pruning techniques designed for ViTs fails in SSMs due to disruption of sequential token positions. To address this, they propose a pruning-aware hidden state alignment method to stabilize the neighborhood of remaining tokens.

**Strengths:**

- The proposed method is well-motivated, with a clear analysis of why existing token pruning techniques fail for SSMs.
- Comprehensive experiments demonstrate efficiency gains across multiple tasks.
- The structure and presentation of the paper are clear and well-organized.

**Weaknesses:**

See the questions below.

**Questions:**

- In Eq.6, how do you handle the accumulation of errors when computing hidden states for multiple consecutive pruned tokens? Does this lead to any instability in longer sequences?
- How sensitive is the performance to the number of pruned tokens? Is there an upper limit to K_i before performance degrades significantly?
- Is there any specific analysis about the pruning rate?

---

> ### Author Rebuttal · Authors · 2024-08-07
>
> We sincerely appreciate the reviewer for recognizing the strengths of our papers and providing valuable feedback. We are happy to address the raised questions as below.
>
> ---
> **Q1. How to handle the consecutive pruned tokens and longer sequences?**
>
> Thank the reviewer for raising this valuable question. Here is a more detailed explanation about how our method handles the accumulation of errors when computing hidden states for multiple consecutive pruned tokens. Our method adopts $h^\prime_{(q_{i})+1}= \mathbf{\overline{A}} h^\prime_{(q_{i})} + \mathbf{\overline{B}} x_{(q_{i})+1} = \mathbf{\overline{A}} h^\prime_{(q_{i})}$ where the token $x_{(q_{i})+1}$ is pruned, which leads to Equation (6).  As shown in Equation 6, if a token is pruned, we do not simply remove its corresponding hidden state during the scan as it leads to substantial accuracy degradation. Instead, its hidden state in the scan can be obtained by using the previous state with one step forward. In this way, the hidden states in SSM corresponding to pruned tokens are aligned with that of the original unpruned tokens to maintain the sequential positions of the original tokens without disrupting their neighbors. Based on the visualization results of Figure A1 in appendix, our method can prune consecutive tokens without additional designs while maintaining good performance.
>
> In our paper, we provide Table 2 and Table A1 to show longer sequence results on object detection with PlainMamba. Where the input size is 1280×800 and 512×2048 respectively. Demonstrating our stability on longer sequences.
>
> ---
> **Q2. How sensitive is the performance to the number of pruned tokens?**
>
> We observe that the performance drops faster after accuracy reaching below 70%. Setting this 70% accuracy as a threshold, when input length is 197:
>
> For ViM-T, there are 60 remaining  tokens after our token pruning, with around 31% FLOPs reduction.
> For ViM-S, there are 36 remaining  tokens, which is around 56% reduction FLOPs.
> We can observe that the larger the model, the more the tokens can be pruned. Also, the upper limit is also closely related to the pruning locations, thus  a better pruning location may  lead to more tokens to be pruned.
> We will include this analysis in the revision.
>
> ---
> **Q3. Is there any specific analysis about the pruning rate?**
>
> Regarding pruning ratio, we set the same pruning ratio (e.g. 0.8) for each location, with a progress pruning manner to prune additional tokens following the pruning ratio for each location. We discuss the pruning locations in Line 263-265.
> The FLOPs reduction can vary due to different pruning locations associated with the number of layers.
>
> We managed to finish evaluating more results with different pruning ratios on ViM-T and ViM-S, we will include more results in the revision.
>
> Table A. Classification results of different ratios on ImageNet-1K.
> |Methods | prune ratio | location | FLOPs(G) |  Top-1 Acc. (%)|
> |-|-|-|-|-|
> | ViM-T | 0.9| [10,20]  |1.38 | 75.6  |
> | ViM-T | 0.8| [10,20]  |1.29 |  75.1|
> | ViM-T | 0.7| [10,20]  |1.21 |  74.5 |
> | ViM-S | 0.9 | [5,10,15,20]  | 4.21 | 79.4 |
> | ViM-S | 0.8 | [5,10,15,20]  |  3.60 | 78.8|
> | ViM-S | 0.7| [5,10,15,20]  | 2.90| 78.1 |

---

> > ### Comment · Reviewer_M2Lk · 2024-08-10
> >
> > I appreciate the author's response. While most of my concerns have been resolved, I'll keep my current rating for now and continue to pay attention to other reviews and ongoing discussions.

---

> > > ### Author Response · Authors · 2024-08-12
> > >
> > > Dear Reviewer,
> > >
> > > Thank you for your prompt response and thoughtful consideration of our rebuttal. We appreciate your acknowledgment of our efforts to provide extensive results and explanations to address your concerns. We sincerely thank you for the constructive comments and positive rating. We will add the constructive suggestions in the final version of our paper.
> > >
> > > Thanks again for your time!
> > >
> > > Warm regards,
> > >
> > > Authors

---

### Official Review · Reviewer_kiGk · 2024-07-12

**Soundness:** 3
**Presentation:** 3
**Contribution:** 3
**Rating:** 5
**Confidence:** 4

**Summary:**

This paper explores the token pruning in vision state space models. It gives the observations that utilizing the token pruning techniques designed for ViTs leads to significant performance drop in SSMs. The main reason is that naive application disrupts the sequential token positions. To solve this, this paper introduces a pruning-aware hidden state alignment method and a token importance evaluation method to guide the token pruning while maintain better performance. Extensive experiments on ImageNet classification and object detection demonstrates the effectiveness of the proposed methods.

**Strengths:**

The main strengths of the paper are:

1、	This paper gives insights about the failure of traditional token-pruning method adopted in ViTs on SSMs. This observation can better guide the token-pruning in SSMs in the future research.

2、	Extensive ablation studies on all relevant components. Apart from the quantitative evaluations, its visual results are also clear.

3、	The paper is well written and easy to read and follow.

**Weaknesses:**

This paper gives interesting observations about the token-pruning in vision state space models. However, I have some concerns about the analysis and the token-pruning methods designed for SSMs.

1、	Fig. 1 illustrates the token computation difference between ViT and ViM. This introduces the large performance gap when adopting the traditional token-pruning techniques. Is there any more quantitative analysis result here to prove the impact of token computation patterns on performance of SSMs, such as randomly exchanging the order of adjacent tokens.

2、	What inspires the design of token importance in Eq(9). I can’t understand the reason of this design. Why adopting value 0 as the clipped threshold? Why does a larger S correspond to the most contextually relevant tokens? The result in Table 3 doesn’t indicate much superiority of the algorithm. In the traditional token-pruning method, the importance of a token should be represented by its’ impact on the loss value.

3、	This paper also introduces the efficient implementation for the SSM scan. The FLOPs and Params are provided to demonstrates the computations reduction. I think we care more about the actual running latency. This information is necessary for us to evaluate the practicality of the methods. The improvement of Throughput in table 4 is not as obvious as that of FLOPs. What caused this?

4、	Some detailed information is missing. Such as what proportions are reduced in different layers? For the SSM model, does the ratio of token reduction in different layers need to follow any principles? I think this can make the article better.

**Questions:**

The observation of this paper is interesting. However, I have major concerns about the novelty of the designed methods. I hope the author can solve my questions in the weaknesses part.

**Limitations:**

Limitations have been discussed in this paper. The efficiency is still limited by the baseline model architecture design.

---

> ### Author Rebuttal · Authors · 2024-08-07
>
> We sincerely appreciate the feedback from the reviewer. We address the raised questions as below.
>
> ---
> **W1. More quantitative analysis result here to prove the impact of token computation patterns on performance of SSMs.**
>
> We would like to thank the reviewer for this valuable suggestion, we conduct additional experiments to randomly shuffle the token positions. For each pruning location of ViM (not all layers), we use the shuffle() function from the random package to randomly exchange the order of tokens. We observe the following performance.
> On ViM-T, the accuracy dropped to 26.35% for zero shot, and 69.2%(-6.9) after finetune.
> On ViM-S, the accuracy dropped to 25.69% for zero shot, and 74.1% (-6.4) after finetune.
> The results indicate that the random token positions lead to much worse performance.
>
> ---
> **W2. Reason behind our design of token importance.**
>
> Currently, SSM is a novel model architecture and to our best knowledge, there are **no previous works** studying the importance of tokens in SSM models, especially for vision SSM models. Therefore, we try to tackle this problem by leveraging previous experience from Transformers. In Transformers, the softmax operation ensures that importance scores are always **positive**. We aimed to maintain a similar property in our approach. That’s why we choose general metrics like $\ell_1$ norm, $\ell_2$ norm and clip at 0 that could provide positive importance score values.
>
> Mamba utilizes implicit attention within its layers. It processes information through SSM layers, allowing tokens to interact and influence each other. By the time tokens reach the output of the SSM layers, they have undergone multiple rounds of implicit interactions and transformations.  Mamba 2[1], has shown that the SSM block is **equivalent** to a form of linear attention. Therefore, the output contains the cumulative effect of these interactions, reflecting how each token has contributed to and been influenced by the overall context of the input. We then aggregate the clipped values across all channels of the output as our token importance score, as discussed in Line 228-231.
>
> Furthermore, the choice of 0 as the clipping threshold serves a role similar to **ReLU** activation:
> 1. It introduces non-linearity, which can help in capturing more complex relationships.
> 2. It improves the stability of backpropagation by preventing negative gradients.
> 3. It could introduce sparsity which we think could benefit the token pruning fine-tuning process. Our experimental results also show that the choice of 0 as the clipping threshold could yield fairly good results.
>
> In Figure 3 of our paper, we provide a visualization of the attention patterns derived from the output, which helps to illustrate the implicit attention mechanism in mamba.
>
> Moreover, deriving token importance directly from the model itself without additional token selection layers or algorithms can be beneficial for specific hardware optimization.
>
> ---
> **Q1. Questions about throughput in table 4.**
>
> Thank the reviewer for pointing this out, the FLOPs of ViM-S was a typo. We have double checked that the accuracy and throughput results in our paper are correct, sorry for the confusion. The "1.21G" typo corresponds to the FLOPS of ViM-T (ratio=0.7),  which has been presented in table of Global rebuttal **A1**. We have reported all the results for better clarity. The updated table is shown below:
>
> |Model| Method | FLOPs(G) |  Top-1 Acc. (%)| Throughput |
> |-|-|-|-|-|
> | ViM-S |Dense|	5.10| 80.5 | 1×|
> | ViM-S |Prune w/o our alignment|3.57| 75.4|  1.30×|
> | ViM-S |Prune w/ our alignment| 3.60|78.8 | 1.27×|
>
> Note that the purpose of this table is to do an ablation study on the effectiveness of the alignment approach, using our token importance pruning method.
> From the results, we can see that our method could improve the throughput by 1.30× with around 30% FLOPS reduction.
>
> ---
> **Q2. Detailed information about pruning ratio and pruned layers.**
>
> Thank the reviewer for pointing out this missing part.
> The pruning location is determined through zero-shot experiments. This allowed us to identify the most effective pruning locations for different model architectures.
>
> The varying number of layers in different models necessitated a flexible approach to partitioning. We found that a uniform partitioning strategy was suboptimal due to these architectural differences.
>
> We would like to clarify that existing works on ViT pruning also have different partitions. For example, Table 3 in DynamicViT paper (TPAMI version) and Section 5 of SPViT paper demonstrate that different scales of ViT models use different pruning locations.
> While the exact pruning locations vary between models, we maintained a **consistent** approach by ensuring **fixed intervals** between pruning stages for each model. This also aligns with methodologies in existing works.
>
> Regarding pruning ratio, we set the same pruning ratio (e.g. 0.8) for each location, with a progress pruning manner, pruning an additional $r$ tokens for each location. We also add additional results of different ratios (e.g. 0.7, 0.9) in **A1**.
>
> We managed to finish evaluating more results with different pruning ratios on ViM-T and ViM-S, we will include more results in the revision.
>
> Table A. Classification results of different ratios on ImageNet-1K.
> |Methods | prune ratio | location | FLOPs(G) |  Top-1 Acc. (%)|
> |-|-|-|-|-|
> | ViM-T | 0.9| [10,20]  |1.38 | 75.6  |
> | ViM-T | 0.8| [10,20]  |1.29 |  75.1|
> | ViM-T | 0.7| [10,20]  |1.21 |  74.5 |
> | ViM-S | 0.9 | [5,10,15,20]  | 4.21 | 79.4 |
> | ViM-S | 0.8 | [5,10,15,20]  |  3.60 | 78.8|
> | ViM-S | 0.7| [5,10,15,20]  | 2.90| 78.1 |

---

> ### Author Response · Authors · 2024-08-12
>
> Dear Reviewer,
>
> Thank you very much for acknowledging our observation and other contributions. Since the discussion will end very soon, we sincerely hope that you have found time to check our detailed response to your previous questions/comments. If you have any further questions, please feel free to let us know. We will try our best to reply to you before the discussion deadline.
>
> Thank you very much,
>
> Authors

---

### Official Review · Reviewer_6myn · 2024-07-13

**Soundness:** 3
**Presentation:** 3
**Contribution:** 2
**Rating:** 4
**Confidence:** 4

**Summary:**

This paper presents a token pruning method for vision state space models. The goal of this paper is to expand the token pruning methods for ViTs to recent SSM-based vision backbones. The authors observed that token pruning will change the computational characteristics of SSMs and lead to significant accuracy drop. To solve this problem, a hidden state alignment method is designed to explicitly adjust the hidden states of SSMs after token pruning. Experiments are conducted on on ImageNet, COCO and ADE20k to show the effectiveness of the method.

**Strengths:**

- The paper is well organized. The analyses in Section 3.2 clearly show the problem that the paper want to solve and help the readers understand the background easily.

- The proposed alignment method and importance metric look reasonable, which are new since they are designed for  SSM-based models.

**Weaknesses:**

- There are still noticeable accuracy drop even after finetuning.  Although there is no previous work on token pruning in SSM-based vision models, many methods are proposed for ViT token pruning and achieved nearly no accuracy drop when reducing around 20% FLOPs. But I find the proposed method will lead to around 0.5 accuracy drop. The results are not that impressive.

- The actual speed-up is not reported. One key advantage of ViT token pruning compared to traditional channel pruning is hardware friend structures after pruning to easily lead to actual speed-up. If the proposed method can also achieve similar results, the method will be much more useful.

**Questions:**

- How to determine the pruning protocol mentioned in line 262-265? It seems that the models are not uniformly partitioned. So it is better to show more details behind the partition strategy used in the experiments.

- How about the results with different pruning ratios? The paper only demonstrate one pruning ratio for each model and the ratios are different for different models.

Overall, I find there are some interesting observations and results presented in the paper. But the experimental studies look a bit weak and the results are not that impressive/useful. Therefore, I would rate this paper as "Borderline reject".

**Limitations:**

The limitations have been discussed in Section 6.

---

> ### Author Rebuttal · Authors · 2024-08-07
>
> We sincerely appreciate the feedback from the reviewer. We address the raised questions as below.
>
> ---
>
> **W1. Accuracy drop after finetuning.**
>
> We appreciate the detailed feedback and would like to address the concern regarding the observed accuracy drop. Our work represents a novel advancement in enhancing the efficiency of SSM-based vision models through tailored token pruning techniques. Unlike ViTs, where existing token pruning methods have been extensively researched and optimized, SSM-based models present unique challenges. As demonstrated in Figure 2 and Table 1 of our main paper, **traditional token pruning methods for ViTs do not perform well on SSMs**.
>
> We would like to highlight that, in comparison to existing designs, our method consistently delivers better accuracy across various SSM-based vision models and scales. For example, our approach surpasses the state-of-the-art (SOTA) token pruning technique for ViTs, EViT, by 3.8% on ViM-T, 4.0% on ViM-S, 2.4% on PlainMamba-L1, 2.7% on PlainMamba-L2, and 2.8% on PlainMamba-L3. This **significant performance improvement** (**over 2.4% across different models**) has been **recognized** and acknowledged by other reviewers (Reviewer M2Lk, TrFe, g8vG). Our method demonstrates superior performance compared to the ViT-based approaches, highlighting the **need for SSM-specific** pruning techniques.
>
> Regarding the 0.5% accuracy drop compared to dense computations on SSM-based vision models, one possible explanation is the inherent difference in computational complexity. SSM-scans achieve linear complexity, which poses greater challenges in attaining high computation reductions compared to the quadratic complexity of the attention mechanism in ViTs, which tends to have more redundant computations. This makes direct comparisons of FLOPs reductions between SSM and ViT models challenging.
>
> On the other hand, the ViM has relatively small and efficient architecture. Smaller models generally have less redundancy, making it more challenging to achieve significant FLOPs reductions without impacting accuracy.
> Our experiments on the larger PlainMamba-L3 model can show a 40% FLOPs reduction with only a 0.6% accuracy drop. This result is better than the performance reported for the transformer model EViT-DeiT-B, which achieved a 35% FLOPs reduction with a 0.5% accuracy degradation (as reported in Table 8 of the EViT paper).
>
> ---
> **W2. Speedup performance.**
>
> We agree that actual speed-up is an important metric to demonstrate the effectiveness of token pruning schemes. Thus, we would like to clarify that we also include the speedup performance of our methods in Table 4 in the main paper. For instance, on PlainMamba-L3, our method with 8.44G FLOPs achieves 1.43$\times$ throughput accelerations than the dense method. Here is the table in our paper,
>
> |Model| Method | FLOPs(G) | Top-1 Acc. (%) | Throughput |
> |-|-|-|-|-|
> | ViM-S |Dense| 5.10| 80.5 | 1×|
> | ViM-S |Prune w/o our alignment|3.57| 75.4| 1.30×|
> | ViM-S |Prune w/ our alignment| 3.60|78.8 | 1.27×|
> | PlainMamba-L3 |Dense| 14.40| 82.3 | 1×|
> | PlainMamba-L3 |Prune w/o our alignment|8.35| 79.3| 1.47×|
> | PlainMamba-L3 |Prune w/ our alignment| 8.44| 81.7 | 1.43×|
>
> ---
> **Q1. How to determine the pruning protocol mentioned in line 262-265.**
>
> Thank the reviewer for raising this valuable question. We will provide more details for comprehensive experimental results. Our partitioning strategy was determined through zero-shot experiments. This allowed us to identify the most effective pruning locations for different model architectures.
>
> The varying number of layers in different models necessitated a flexible approach to partitioning. We found that a uniform partitioning strategy was suboptimal due to these architectural differences.
> Existing works on ViT pruning also have different partitions. For example, Table 3 in the DynamicViT [1] paper and Section 5 of SPViT [2] paper demonstrate that different scales of ViT models use different pruning locations.
>
> While the exact pruning locations vary between models, we maintained a consistent approach by **ensuring fixed intervals** and **same pruning rate** between pruning stages for each model. This also aligns with methodologies in existing works.
> In the paper, we constantly set the pruning ratio to 0.8 for each location. The FLOPs reduction difference is due to the difference in model size related to the pruning location of layer index as stated in Line 262-265.
>
> We also include more detailed information about Pruning ratio and pruned layers in Global rebuttal A1.
>
> [1] Dynamic Spatial Sparsification for Efficient Vision Transformers and Convolutional Neural Networks. TPAMI
>
> [2] SPViT: Enabling Faster Vision Transformers via Soft Token Pruning, ECCV 2022
>
> ---
> **Q2. Results with different pruning ratios.**
>
> In the paper, we constantly set the pruning ratio to 0.8 for each location. The FLOPs reduction difference is due to the difference in model size related to the pruning location of layer index as stated in Line 262-265. We managed to finish evaluating more results with different pruning ratios on ViM-T and ViM-S, we will include more results in the revision.
>
> Table A. Classification results of different ratios on ImageNet-1K.
> |Methods | prune ratio | location | FLOPs(G) |  Top-1 Acc. (%)|
> |-|-|-|-|-|
> | ViM-T | 0.9| [10,20]  |1.38 | 75.6  |
> | ViM-T | 0.8| [10,20]  |1.29 |  75.1|
> | ViM-T | 0.7| [10,20]  |1.21 |  74.5 |
> | ViM-S | 0.9 | [5,10,15,20]  | 4.21 | 79.4 |
> | ViM-S | 0.8 | [5,10,15,20]  |  3.60 | 78.8|
> | ViM-S | 0.7| [5,10,15,20]  | 2.90| 78.1 |
>
> ---
> Overall, we take the **first step** toward accelerating vision SSM models with token-based pruning. We conduct a comprehensive analysis of SSM-based blocks to identify the **failure reason**, as well as provide more **insights** for the SSM scan mechanism in vision tasks, shedding lights on future research on SSM-based vision models.

---

> ### Author Response · Authors · 2024-08-12
>
> Dear Reviewer,
>
> Thank you very much for acknowledging our observation and other contributions. Since the discussion will end very soon, we sincerely hope that you have found time to check our detailed response to your previous questions/comments. If you have any further questions, please feel free to let us know. We will try our best to reply to you before the discussion deadline.
>
> Thank you very much,
>
> Authors

---

> > ### Comment · Reviewer_6myn · 2024-08-13
> >
> > Thanks for the detailed feedback. It is good to see that the method can also lead to actual speed-up after pruning. However, my concerns are still not fully addressed:
> >
> > - The actual speed-up of the proposed method seems less ideal than that of the ViT pruning method like DynamicViT. I am still concerned about the value of studying token pruning for SSM-based models.
> >
> > - EViT is a token pruning baseline published in 2022. Recent methods like AViT [r1] and STViT [r2] can achieve no accuracy drop after pruning. The results presented in the paper are relatively weak.
> >
> > [r1] AdaViT: Adaptive Tokens for Efficient Vision Transformer, CVPR 2022.
> > [r2] Making Vision Transformers Efficient from A Token Sparsification View, CVPR 2023.
> >
> > Overall, I would keep my original rating.

---

> ### Author Response · Authors · 2024-08-14
>
> **We sincerely appreciate the feedback from the reviewer. We address the newly raised questions below.**
>
> ---
> **Q1. The actual speed-up of the proposed method seems less ideal than that of the ViT pruning method like DynamicViT. I am still concerned about the value of studying token pruning for SSM-based models.**
>
> Token pruning is crucial for enhancing efficiency, especially with long sequences (please refer to our answer to **Q1** of Reviewer **M2Lk**). By reducing the number of tokens processed, we can significantly decrease computational cost and memory usage, thereby improving the scalability of SSMs for real-world applications. In this work, we introduce a general token pruning method specifically designed for SSM-based vision models. This is the **first successfully** token pruning work to **effectively** handle this problem on SSMs with **huge performance improvement**.
>
> Given the limited time (less than one day), we conducted an acceleration comparison between our method and DynamicViT under the same pruning ratio on SSMs. The results are as follows:
>
> |Model| Method | FLOPs(G)| Throughput |
> |-|-|-|-|
> | ViM-S |Ours| 3.60 | 1.27×|
> | ViM-S |DynamicViT| 3.56 | 1.30×|
> | PlainMamba-L3 |Ours|8.44|  1.43×|
> | PlainMamba-L3 |DynamicViT| 8.37 | 1.46×|
>
> As shown, there is minimal difference in acceleration under the same pruning ratio when compared to methods like DynamicViT on SSMs. However, our method provides **much higher** accuracy compared to ViT methods. For example, recent works such as ToMe [1] and LTMP [2], which we discussed in our response to **Reviewer TrFe**, highlight that our method achieves **significant performance improvements** over token pruning methods in ViT. The results are as follows:
>
> |Methods | FLOPs(G) |  Top-1 Acc. (%)|
> |------------|-----|-----|
> | ViM-T  |1.50| 76.1 |
> | ViM-T-ToMe |	1.28|71.6|
> | ViM-T-LTMP |	1.29|72.2|
> | ViM-T-EViT | 1.28|71.3|
> | ViM-T-Ours | 1.29 |75.1|
>
> ---
> **Q2. EViT is a token pruning baseline published in 2022. Recent methods like AViT [r1] and STViT [r2] can achieve no accuracy drop after pruning. The results presented in the paper are relatively weak.**
>
> While different token pruning methods in ViT may prune tokens without reducing accuracy, they all face the **same challenges** when applied to SSMs. In SSM, the relationships between tokens are **different** from those in ViT. SSMs handle sequential data, where the dynamic characteristics and contextual dependencies of each token determine its importance. Simple pruning might disrupt the sequence's integrity or the transmission of crucial information, leading to **significant** performance **degradation**. We would like to highlight that our work is the first successful application of token pruning to SSMs, resulting in substantial performance improvements.
>
> Due to the limited time (less than one day for applying AViT and STViT), we summarize the more recent work like ToMe [1] and LTMP [2] from our answer to **reviewer TrFe**. We will add these results into our revised version. We implemented ToMe[1] as well as LTMP [2] for our SSM-based models to provide a comprehensive comparison with state-of-the-art techniques. As demonstrated in the following table, our method can outperform all baselines with non-marginal improvements.
>
> |Methods | FLOPs(G) |  Top-1 Acc. (%)|
> |------------|-----|-----|
> | ViM-T  |1.50| 76.1 |
> | ViM-T-ToMe |	1.28|71.6|
> | ViM-T-LTMP |	1.29|72.2|
> | ViM-T-EViT | 1.28|71.3|
> | ViM-T-Ours | 1.29 |75.1|
>
> **Overall, we would like to emphases that regardless of the ViT token pruning method, if it does not address the specific characteristics of SSMs, it will not achieve the same level of effectiveness in SSMs. Therefore, it is unreasonable to compare the accuracy drop level on ViT with it on SSMs.**
>
> [1] Token Merging: Your ViT But Faster, ICLR 2023
>
> [2] Learned Thresholds Token Merging and Pruning for Vision Transformers, TMLR 2023

---

> > ### Comment · Reviewer_6myn · 2024-08-14
> >
> > Thanks for your new and detailed results. I agree that the proposed method can clearly improve existing methods that are not designed for SSM-based models, as I mentioned in "Strengths". However, I think the authors may misunderstand my core concern. I think there is still no clear evidence that SSM-based models are better than ViTs on visual tasks, and ViTs is still the first choice for core application scenarios of visual backbones including visual understanding (like detection, segmentation), and multimodal modeling (like CLIP, MLLMs). Therefore, in my opinion, only if achieving results comparable to ViTs' efficiency/performance can make a solid and impressive impact, simply beating some weak baselines may not fully demonstrate the value of studying this problem.

---

> ### Author Response · Authors · 2024-08-14
>
> Thank you for your detailed feedback and for acknowledging the strengths of our work. We understand your core concern regarding the comparative performance of SSM-based models and ViTs on vision tasks, and we'd like to address this more explicitly.
>
> 1. **SSMs as an Emerging Foundation Model:** While it is true that ViTs have demonstrated strong performance across a variety of visual tasks, SSMs represent a new and rapidly evolving class of models. Given their emergent nature, it may be premature to conclude that ViTs are definitively superior to SSMs. In fact, SSMs have already shown **remarkable performance** in several vision tasks, indicating their potential to be **strong competitors** or even **successors** to ViTs as research in this area progresses.
>
> 2. **Our Baselines are Not Weak:** We want to clarify that the baselines we compare against are not weak; they represent the state of the art in the relevant categories. Our method **significantly outperforms** these **SOTA** methods, demonstrating the practical value and effectiveness of our approach within the current landscape of vision SSM models.
>
> 3. **Contributions Beyond Baseline Comparisons:** Our work goes beyond simply applying token pruning to a vision SSM model. We provide critical insights into the design of future vision SSM models, particularly regarding the handling of **token continuity** and **positional information**. These contributions are **foundational** and offer **valuable guidance** for the continued development of vision SSM models. It is important to recognize that early-stage work in emergent fields often serves as a **foundation stone**, even if it does not immediately surpass established benchmarks in ViTs-based models. The insights we offer pave the way for future innovations and improvements in the design of SSMs for vision tasks.

---

### Author Rebuttal · Authors · 2024-08-07

We would like to express our gratitude to the reviewers for their positive comments and constructive feedback on the paper. We sincerely appreciate the reviewers for acknowledging **our motivation** is clear and reasonable (Reviewer TrFe), with interesting observations that give insights to better guide the token-pruning in SSMs in the future research (Reviewer 6myn, kiGk), **our method** is new, resonable (Reviewer 6myn), well-motivated (Reviewer M2Lk), and effective with good performance (Reviewer g8vG), **our experiments and results** are extensive, comprehensive, and impressive with both quantitive and visual evaluations (Reviewer kiGk, M2Lk, TrFe, g8vG),  and **our paper** is well-organized, well-written, and easy to read and follow (Reviewer 6myn, kiGk, M2Lk, TrFe).

---

**A1. Detailed information about Pruning ratio and pruned layers.**

In the paper, we constantly set the pruning ratio to 0.8 for each location. The flops reduction difference is due to the difference in model size related to the pruning location of layer index as stated in Line 262-265. We will include more results in the revision. Here are the results about pruning ratio and pruned layers,

Table A. Classification results of different ratios on ImageNet-1K.
|Methods | prune ratio | location | FLOPs(G) |  Top-1 Acc. (%)|
|-|-|-|-|-|
| ViM-T | 0.9| [10,20]  |1.38 | 75.6  |
| ViM-T | 0.8| [10,20]  |1.29 |  75.1|
| ViM-T | 0.7| [10,20]  |1.21 |  74.5 |
| ViM-S | 0.9 | [5,10,15,20]  | 4.21 | 79.4 |
| ViM-S | 0.8 | [5,10,15,20]  |  3.60 | 78.8|
| ViM-S | 0.7| [5,10,15,20]  | 2.90| 78.1 |


**A2. Explanation of our design of Importance Evaluation.**

Currently, SSM is a novel model architecture and to our best knowledge, there are **no previous works** studying the importance of tokens in SSM models, especially for vision SSM models. Therefore, we try to tackle this problem by leveraging previous experience from Transformers. In Transformers, the softmax operation ensures that importance scores are always **positive**. We aimed to maintain a similar property in our approach. That’s why we choose general metrics like $\ell_1$ norm, $\ell_2$ norm and clip at 0 that could provide positive importance score values.

Mamba utilizes implicit attention within its layers. It processes information through SSM layers, allowing tokens to interact and influence each other. By the time tokens reach the output of the SSM layers, they have undergone multiple rounds of implicit interactions and transformations.  Mamba 2[1], has shown that the SSM block is **equivalent** to a form of linear attention. Therefore, the output contains the cumulative effect of these interactions, reflecting how each token has contributed to and been influenced by the overall context of the input. We then aggregate the clipped values across all channels of the output as our token importance score, as discussed in Line 228-231.

Furthermore, the choice of 0 as the clipping threshold serves a role similar to **ReLU** activation:
1. It introduces non-linearity, which can help in capturing more complex relationships.
2. It improves the stability of backpropagation by preventing negative gradients.
3. It could introduce sparsity which we think could benefit the token pruning fine-tuning process. Our experimental results also show that the choice of 0 as the clipping threshold could yield fairly good results.

In Figure 3 of our paper, we provide a visualization of the attention patterns derived from the output, which helps to illustrate the implicit attention mechanism in mamba.

Moreover, deriving token importance directly from the model itself without additional token selection layers or algorithms can be beneficial for specific hardware optimization.

[1] Transformers are SSMs: Generalized Models and Efficient Algorithms Through Structured State Space Duality,

---

### Decision · Program_Chairs · 2024-09-25

**Decision:**

Accept (poster)

**Comment:**

This paper proposes a novel technique to prune State-State Models (SSMs) tokens for vision tasks based on insights specific to SSMs. Experiments are conducted on ImageNet-1K, COCO 2017, and ADE20K datasets.

Upon the rebuttal and the authors-reviewers discussion, the reviews were slightly diverging, and three were borderline: 1x borderline reject, 2x borderline accept, and 1x weak accept.

The AC summarizes these main aspects:

(1) The paper is clear.

(2) The proposed model compression by token pruning for SSMs-based vision models is novel. This virtue mainly stems from considering the recent hot trend of SSMs. Although token pruning is well known, the paper provides insights into applying the technique to SSMs, which are informative and not trivial.

(3) The performance improvement is satisfactory. The majority of reviewers and the AC reckon the compromise between computation reduction and accuracy is analyzed thoroughly, demonstrating the proposed technique's quality. Two reviewers have demanded comparisons against more baselines other than the comparison with EViT in the submitted paper. The authors could not provide a comparison with AViT [cvpr'22] and STViT [cvpr'23], which one reviewer suggested, but they provided a comparison with ToMe (ICLR'23) and LTMP (TMLR'23). One of the reviewers deems this is sufficient, and the AC seconds this statement but encourages the authors to complete the performance evaluation and consider those baselines if they can manage before the camera-ready submission deadline.

Overall, the AC stands with the favorable opinion of most reviewers and recommends this paper for acceptance.